# A Patient-Specific Fracture Risk Assessment Tool for Femoral Bone Metastases: Using the Bone Strength (BOS) Score in Clinical Practice

**DOI:** 10.3390/cancers14235904

**Published:** 2022-11-29

**Authors:** Florieke Eggermont, Yvette van der Linden, Nico Verdonschot, Edwin Dierselhuis, Steven Ligthert, Thom Bitter, Paulien Westhoff, Esther Tanck

**Affiliations:** 1Orthopaedic Research Laboratory, Radboud Institute for Health Sciences, Radboud University Medical Center, 6525 GA Nijmegen, The Netherlands; 2Department of Radiotherapy, Leiden University Medical Center, 2333 ZA Leiden, The Netherlands; 3Netherlands Comprehensive Cancer Organisation (IKNL), 3511 DT Utrecht, The Netherlands; 4Laboratory of Biomechanical Engineering, University of Twente, 7522 NB Enschede, The Netherlands; 5Department of Orthopedics, Radboud Institute for Health Sciences, Radboud University Medical Center, 6525 GA Nijmegen, The Netherlands; 6Department of Radiation Oncology, Radboud Institute for Health Sciences, Radboud University Medical Center, 6525 GA Nijmegen, The Netherlands

**Keywords:** fracture risk, finite element model, bone strength, femoral bone metastases

## Abstract

**Simple Summary:**

Patients with femoral metastases are at risk of developing a fracture, whose prevention is important for maintaining mobility and, hence, patients’ quality of life. The BOne Strength (BOS) score is a computed tomography (CT)-based patient-specific computer model that objectively calculates the bone strength of femurs with metastases. It was developed to aid patients and their treating physicians in selecting appropriate treatment options, either radiotherapy in low-risk femurs or elective stabilizing surgery in high-risk femurs. In this pilot study, the added clinical value of the BOS score in treatment-related decision making was assessed. The study showed that the BOS score is a promising and objective tool to assess fracture risk in femoral bone metastases and aids physicians and patients in making a more informed decision regarding the most appropriate treatment.

**Abstract:**

Patients with femoral metastases are at risk of fracturing bones. It is important to prevent fractures in order to maintain mobility and quality of life. The BOne Strength (BOS) score is based on a computed tomography (CT)-based patient-specific finite element (FE) computer model that objectively calculates bone strength. In this pilot study, the added clinical value of the BOS score towards treatment-related decision making was assessed. In December 2019, the BOS score was implemented in four radiotherapy centers. The BOS scores and fracture risks of individual patients were calculated and returned to the physician to assist in treatment decisions. The physicians filled out a questionnaire, which was qualitatively analyzed. A follow-up to identify fractures and/or death was performed after six months. Until June 2021, 42 BOS scores were delivered (20 high, 9 moderate, and 13 low fracture risk). In 48%, the BOS score led to an adaptation of treatment plans. Physicians indicated that the BOS score provided objective insight into fracture risk, was reassuring for physicians and patients, and improved multidisciplinary discussions and shared decision making. In conclusion, the BOS score is an objective tool to assess fracture risk in femoral bone metastases and aids physicians and patients in making a more informed decision regarding the most appropriate treatment.

## 1. Introduction

The incidence of cancer increases every year, and with that, the number of patients with metastasized cancer also increases [1]. Bone is a common site for metastases [2]. Bone metastases can have different appearances. Osteolytic metastases result from disproportionate bone resorption by osteoclasts, whereas osteoblastic metastases are caused by excessive bone formation [2]. Mixed-type metastases, where osteoblastic and osteolytic lesions coexist, are also common. Patients with bone metastases have a certain risk of developing a pathological fracture. Pathological fractures in the femur have a large effect on a patient’s mobility and self-care. Moreover, having a sudden pathological fracture evidently causes anxiety and stress for the patient. As a result, the quality of life is considerably affected. Pathological fractures have also been associated with decreased survival [3,4]. Compared to the prophylactic surgery of impending fractures, the surgical treatment of pathological fractures has been shown to lead to greater morbidity as well as longer hospital stays and higher costs [4,5,6].

Therefore, the effective treatment of patients with femoral bone metastases is dependent on the risk of a pathological fracture. Patients with an expectedly low fracture risk are treated conservatively with radiotherapy to relieve pain, whereas patients with an expectedly high fracture risk are considered for prophylactic stabilizing surgery [7]. If a patient has a high fracture risk but an insufficient general health status to undergo surgery, or if surgery is refused by the patient, the treatment usually consists of a higher dose of radiotherapy divided over multiple fractions to induce remineralization to strengthen the affected femur [8]. Thus, the key element of good clinical care is fracture risk assessment.

For clinicians, it is difficult to estimate fracture risk based on radiological imaging. Risk assessment tools that were developed to aid fracture risk prediction and that are currently used in clinical practice, such as the Mirels’ score, Harrington’s criteria, or 30 mm axial cortical involvement, mostly overestimate the risk of fracture and, therefore, lead to surgical overtreatment [7,8,9,10,11,12,13,14,15] and insufficient care.

To improve femoral fracture risk assessment and aid patients and their treating physician, such as radiation oncologists and orthopedic surgeons, in selecting appropriate treatment options, the BOne Strength (BOS) score was developed [16,17,18,19]. The BOS score is an easy-to-use score representing the bone strength of a femur, which is objectively calculated by a patient-specific finite element (FE) computer model. In a previous prospective cohort study, it was shown that the BOS score improved fracture risk assessments in comparison to the currently used axial cortical involvement of the metastases (sensitivity of 100% vs. 86%, specificity of 74% vs. 42%, positive predictive value of 39% vs. 19%, and negative predictive value of 100% vs. 95%, respectively) [19].

As a result, in December 2019, we started a pilot study for clinical implementation in four institutes that also participated in our previous patient studies [18,19]. In the current study, the goal is to obtain an initial impression of the added clinical value of the BOS score on treatment-related decision making.

## 2. Methods

### 2.1. Study Design

In December 2019, the pilot study was initiated at the Radboud university medical center, Leiden University Medical Center, Institute Verbeeten Tilburg, and Radiotherapeutic Institute Friesland. Ethical approval was obtained by an accredited medical research ethics committee (Commissie Medische Ethiek Leids Universitair Medisch Centrum (P17.308)) as well as the local ethics committees in all participating centers (Commissie Mensgebonden Onderzoek regio Arnhem-Nijmegen (2019-5494); Medisch-Etische Toetsingscommissie Leiden Den Haag Delft (N19.089), Local Research Committee Verbeeten; Local Research Committee RIF). Patients with predominantly osteolytic femoral bone metastases who visited the radiotherapy departments were asked to participate in the study and to sign informed consent. Patients were included if they were affected with femoral bone metastases confirmed by diagnostic imaging and caused by a histologically or cytologically proven solid tumor. Patients were excluded if they already showed evidence of a pathological fracture, had metal devices implanted in the femur or contralateral femur, or had predominantly osteoblastic metastases.

### 2.2. Requesting a BOS Score

To request a BOS score, an application form was filled in by the physician (mainly radiation oncologists) and was sent to the central study site, the Orthopaedic Research Lab (Nijmegen, The Netherlands) of the Radboud university medical center. This form included general information about the patient (age, weight, primary tumor, location and type of bone metastases, pain score, and Karnofsky performance score (KPS)) and for which femur the BOS score was requested. Patients filled in the EQ-5D-3L [20] together with the physician to assess the self-rated quality of life. Physicians were also asked to assess the initial fracture risk assessment beforehand to compare with the fracture risk based on the BOS score. Additionally, patients underwent a radiotherapy-planning computed tomography (CT) scan with protocolled settings that were validated to calculate the BOS score: 120 kVp, variable mA dependent on the patient’s size, slice thickness of 3 mm, pitch 1.5, spiral and standard bone reconstruction, field of view (FOV) 480 mm, and in-plane resolution of 0.9375 mm [17,18,19]. According to the protocol, a solid calibration phantom containing known calcium equivalent densities (Image Analysis, Columbia, KY, USA) was scanned together with the patient [17,18,19]. The CT scan was sent to the Orthopaedic Research Lab via a secured DICOM server (used for Radboudumc) or via a web-based application that allowed authenticated users to securely and easily send arbitrarily large files to other users (SURFfilesender; used for other institutes).

### 2.3. FE Model and BOS Score Calculation

At the Orthopaedic Research Lab, the BOS score was calculated. CT scan settings as well as the appearance of the metastases (osteolytic, osteoblastic, or mixed) were checked by the BOS technician. Subsequently, the FE model was generated as described previously [18,19,21]. During the past years, the workflow was updated a number of times. Herein, the current workflow is shortly described. First, the Hounsfield Units (HU) in the CT scan were calibrated to calcium equivalent densities (i.e., a measure of bone density). Subsequently, the femur geometry was segmented from the CT scan using a convolutional neural network (https://grand-challenge.org/algorithms/femur-segmentation-in-ct/ (accessed on 4 August 2021)) and converted to a solid mesh of tetrahedral elements (Altair SimLab (Altair, Troy, MI, USA), MATLAB iso2mesh toolbox version 1.8.0 [22], and Patran 2021, MSC Software Corporation, Santa Ana, CA, USA). Each element was assigned its own bone density [23,24]. The mechanical behavior of each element was determined based on the bone densities [25]; hence, simply put, elements with a lower bone density will have a lower strength and stiffness. In this way, osteolytic metastases, which have lower bone density, will automatically have lower strength and stiffness in comparison to normal bone. The femur was aligned to mimic stance by aligning the knee center with the femoral head center. Subsequently, the proximal half of the femur was selected to be included in the FE model (Marc Mentat 2021.1, MSC Software Corporation, Santa Ana, CA, USA). In case the metastasis was in the distal part of the femur, the selected part was extended to include the metastasis. In an FE simulation (MSC.MARC 2021, MSC Software Corporation, Santa Ana, CA, USA), the femur was loaded until fracture. The strength of the femur was determined by the maximum total reaction force (in N). The BOS score was calculated by dividing the strength of the femur by the body weight of the patient (in N). The weakest location of the femur was defined by the elements that had deformed plastically at the moment of maximal total reaction force.

### 2.4. BOS Score Report

In the report (see Appendix A for an example report), the BOS score was visualized in relation to all BOS scores in the database. These are from previous patients for whom it is known whether or not they developed a fracture within six months after the CT scan. In a previous study [18], a threshold of 7.5 for differentiating between femurs with a high and low fracture risk was determined. A moderate fracture risk was defined as a BOS score between 7.5 and 8.5. Using this threshold, a fracture risk estimation was included in the report, together with the positive and negative predictive values for interpretation purposes. Additionally, the BOS report comprised a figure that showed the weakest location of the bone according to the FE model. The report was sent back to the requesting physician within a maximum of three working days.

After the BOS score was delivered, physicians filled out a questionnaire comprising one question on the eventual treatment decision, and the following five questions on their experience with the use of the BOS score (1 yes/no and 4 open-ended questions):Did you use the BOS score (yes or no)?Are you satisfied with the BOS score, and why?Was the BOS score of added value for the treatment decision, and why?Was the patient satisfied regarding the use of the BOS score, and why?Do you have additional points for improvement or other comments, for example, regarding the application form or the BOS report?

It should be mentioned that it was not mandatory to complete the open-ended questions.

Six months after inclusion, follow-up was performed to determine fracture and/or death using the hospital’s electronic patient files. In case information was missing in the electronic patient files, the general practitioner was consulted.

### 2.5. Analysis

The results of all patients included until June 2021 are reported. Time between BOS score request and delivery of report were registered. Changes in treatment plans were estimated by comparing the physician’s initial fracture risk estimation and the BOS score with the final treatment decision. Standard treatment for low-risk patients is a single dose of 8 Gy radiotherapy, and for high-risk patients, a higher dose divided over multiple fractions or prophylactic surgery. To assess the experience with the BOS scores and the value of the BOS score, the answers given by the physicians on the open-ended questions were analyzed qualitatively by applying inductive coding using thematic analysis [26]. The frequency of identified themes was then analyzed using descriptive statistics. Coding was performed by one researcher (FE). The final set of codes was shared with another research team member for consensus (YL) so as to obtain a definitive set of identified themes.

## 3. Results

### 3.1. Patients

Between December 2019 and June 2021, the first 42 BOS reports were delivered for 39 patients (for three patients, a bilateral BOS score was requested): 22 patients (24 femurs) were included at the Radboudumc, 10 patients (11 femurs) were included at LUMC, and 7 patients (7 femurs) were included at Institute Verbeeten. BOS scores were requested by eighteen different physicians, of which ten requested more than one BOS score. Patient characteristics can be found in Table 1. Three femurs (all high fracture risk according to the BOS score) fractured within the six-month follow-up (7%), two of them had fractured shortly prior to the scheduled elective surgery. Thirteen patients (33%) died within the 6-month follow-up period.

### 3.2. Effect on Treatment Decisions

To estimate the effect of the BOS score on the eventual treatment decisions, for each femur (*n* = 42), the initial fracture risk estimation by the physician was compared with the fracture risk based on the BOS score and with the final treatment delivered (Table 2).

For five femurs, both the initial estimation by the physician and the BOS score indicated a high fracture risk. These five patients were all scheduled for elective surgery, and in two patients, a pathological fracture occurred shortly prior to the scheduled surgery. In one case, the patient refused surgery but changed her mind after the BOS score indicated a high fracture risk (see Case A).

For six femurs, both the initial estimation by the physician and the BOS score indicated a low fracture risk (see Case B for an example). One patient was treated with multiple fractions of radiotherapy because of pain complaints, four patients received a single dose, and one patient was not treated at all.

For 31 femurs, a discrepancy existed between the physician’s estimation and the BOS score. For 22 femurs, the initial fracture risk was assessed as low, but the BOS score indicated a high risk (*n* = 15) or moderate risk (*n* = 7). In 18 of these cases, the treatment plan was adapted based on the BOS score (1x elective surgery and 17x higher dose of radiotherapy; see Case C for an example). One patient who received multiple fractions of radiotherapy developed a fracture.

In seven femurs, the initial fracture risk was estimated as high, but the BOS score indicated a low fracture risk. In two cases, the patients were treated with a single fraction of radiotherapy as one would expect in the case of a low fracture risk [7]. The other five patients were treated with a higher dose of radiotherapy divided over multiple fractions. In most cases, it was indicated that the low risk indicated by the BOS score was a sufficient reason to refrain from surgery.

For two femurs, there was a high initial fracture risk and a moderate risk according to the BOS score. One patient refrained from treatment because the patient was not in pain. The other patient was treated with multiple fractions radiotherapy with the aim to improve bone strength and thereby avoid a fracture.

### 3.3. Experiences of Physicians with the Use of the BOS Score

Fourteen of the forty-two BOS reports were delivered the same day (33%), twenty-three were delivered the following day (55%), and five were delivered on the second day (12%). In general, physicians mentioned that they were very satisfied with the short delivery times. Physicians indicated that they used the BOS score in 40 out of 42 cases (95%). In two cases, the BOS score was not used; one patient was too fragile for surgery and the other refused surgery prior to obtaining the BOS score.

To investigate the first experiences of physicians with the BOS scores, common themes were identified from the narrative answers to the open-ended questions (Table 3). Four themes were identified: “clarity of the BOS score”, “effect on decision of treatment”, “reassurance”, and “shared decision making”.

Regarding the “clarity of the BOS score”, it was stated 16 times that the BOS score was a clear, objective score and 11 times that it gave insight into fracture risk. In three cases, it was mentioned that when a BOS score assigned a moderate fracture risk, it was difficult to decide what the consequence for the actual treatment was.

Regarding the “effect on decision of treatment”, physicians stated 17 times that the BOS score had a decisive effect on the treatment decision. It was mentioned 10 times that treatment decisions were better substantiated. Another positive aspect of the BOS score was that radiation oncologists often consulted orthopedic surgeons to discuss the BOS score and the possibility of prophylactic stabilization, thereby facilitating a multidisciplinary consultation (mentioned eight times in the answers of the questionnaires).

Another theme was “reassurance”. If the BOS score was in correspondence with the clinical fracture risk estimation, it gave both physicians (16 times) and patients (6 times) additional confirmation that the most appropriate treatment decision was made.

Regarding the theme “shared decision making”, it was mentioned 10 times that the BOS score helped to open the conversation between the physician and patient.

**Case A:** A female patient (age 77 years, salivary duct carcinoma, KPS 60) was referred to the orthopedic surgeon because of a large painful lytic metastasis in her femur (pain score 8 on a scale of 0 to 10). The orthopedic surgeon suggested planning a prophylactic stabilizing surgery. However, the patient hesitated; hence, a BOS score was requested. The BOS score indicated a high fracture risk, which was discussed with the patient by the orthopedic surgeon and the radiation oncologist. It convinced the patient to undergo the prophylactic stabilizing surgery. One day prior to the scheduled surgery, she fractured her femur while stepping out of her bed. The location of the fracture was similar to the weakest location of the bone indicated by the BOS score. The fracture was treated during surgery. Two months later, she passed away. **Case B:** A female patient (age 76 years, breast cancer, KPS 80) visited the radiation oncologist because of a painful lytic metastasis in the femoral head (pain score 5 on a scale of 0 to 10). The radiation oncologist estimated the fracture risk as low and requested a BOS score, which also indicated a low fracture risk. The patient was treated with 1 × 8 Gy radiotherapy. After a follow-up of six months, the patient had not developed a fracture and was still alive. **Case C:** A female patient (63 years, breast cancer, KPS 80) visited the radiation oncologist because of a painful metastasis in the femoral shaft (pain score 9 on a scale of 0 to 10). The metastases had a mixed appearance but were predominantly lytic. The radiation oncologists initially estimated the fracture risk as low, and the initial plan was to treat the patient with a radiotherapy dose of 1 × 8 Gy. However, the BOS score indicated a moderate risk of fracture. The patient was, therefore, treated with 2 × 8 Gy radiotherapy with the aim to induce remineralization of the bone. After six months, the patient had not developed a fracture and was still alive. 
**BOS score of Femur A depicted relative to patients in the BOS database**


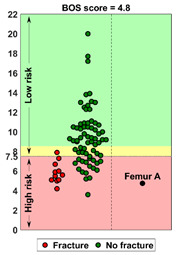


**BOS score of Femur B depicted relative to patients in the BOS database**


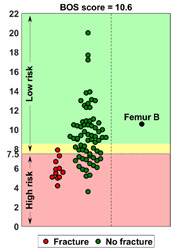


**BOS score of Femur C depicted relative to patients in the BOS database**


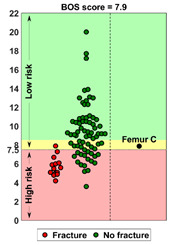

**Weakest location of the bone**ATTENTION: Fracture localization by experimental fracturing in computer model does not necessarily coincide with metastatic location
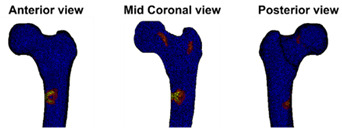
**Weakest location of the bone**ATTENTION: Fracture localization by experimental fracturing in computer model does not necessarily coincide with metastatic location
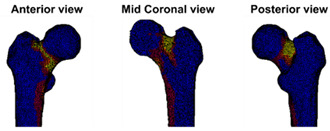
**Weakest location of the bone**ATTENTION: Fracture localization by experimental fracturing in computer model does not necessarily coincide with metastatic location
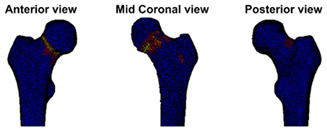


## 4. Discussion

In this study, the added clinical value of the BOS score (obtained from a patient-specific FE model) towards treatment-related decision making in patients with femoral bone metastases was shown. The results of this pilot study indicated that the BOS score is of added value for physicians such as radiation oncologists and orthopedic surgeons because it is a clear score that provides objective insight into fracture risk. Some treatment plans were adapted because of the fracture risk assessment of the BOS score. For patients as well as physicians, it was reassuring that an objective computer model calculated the risk of fracture. Another benefit of the BOS score was that it seemed to result in greater multidisciplinary consultation. In addition, in some cases, the BOS score opened the conversation between the patient and physician, which probably improved shared decision making.

The diagnostic accuracy of the BOS score appears to be higher compared to other methods, such as 30 mm axial cortical involvement, which is used in Dutch clinical practice [19] (see Appendix A for current database). Physicians indicated that most of the BOS scores were used and useful to determine the treatment decision. If the BOS score estimated a higher risk than the physician estimated, radiotherapy doses were often increased with the aim to induce remineralization of the bone. In a few cases, the BOS score resulted in lower radiotherapy doses, mainly in cases in which both the physician and the BOS score estimated a low fracture risk.

To our knowledge, the BOS score is currently the only FE-based model that is used in clinical practice for the fracture risk assessment of patients with femoral bone metastases. For patients with osteoporosis, two research groups implemented FE models on a small scale. Keaveny et al. [27] have developed VirtuOst Biomechanical Computed Tomography analysis (BCT), a tool that uses an FE model to assess fracture risk in patients with osteoporosis. Based on an FE model simulating a side-fall load and the patient’s BMD, the patient is assessed as high-risk, increased risk, or no increased risk, which seems to be a comparable stratification to that of the BOS score. Benemerito et al. [28] have developed and thoroughly described a workflow for fracture risk prediction using FE models for patients with osteoporosis. They generated a patient-specific FE model and applied 28 different load simulations, which will possibly give a more detailed insight into the fracture risk compared to the single loading condition that is used for the BOS score. They subsequently reported the minimum and maximum femur strengths in Newtons under stance and side-fall loads to the medical specialist. It is, however, very difficult to assign a certain value of strength in Newtons with respect to high or low fracture risk, which may make their report difficult to interpret for physicians. We have tried to create a BOS report that is very insightful for medical specialists by including the BOS score together with the fracture risk in relation to the entire database of patients, for whom it is known whether or not they developed a femoral fracture six months after the BOS score was calculated. Additionally, the PPV and NPV of the BOS score are clearly reported for interpretation purposes (see Appendix A for an example of a BOS report). Physicians confirmed that the BOS report was clearly interpretable.

A multidisciplinary approach for the management of bone metastases has shown to improve quality of life of patients [29]. Although a multidisciplinary consultation between radiation oncologists and orthopedic surgeons is advised when determining the treatment for patients with bone metastases [30], we noticed that such consultations were not always organized in our clinical practice. We showed that the BOS score can facilitate a more multidisciplinary fracture risk assessment, as radiation oncologists and orthopedic surgeons together discussed the BOS score and possible treatments. A disadvantage is that treatment may be postponed if the BOS score obliges the further discussion the fracture risk between radiation oncologists and orthopedic surgeons.

Furthermore, shared decision making is an important part of care for patients with advanced cancer. Previous studies have shown that shared decision making ensures that the patient is more informed about the treatment and its harms and benefits [31]. Additionally, shared decision making is known to increase satisfaction [31] and improve emotional functioning [32], the experienced quality of care [33], and, possibly, quality of life [34]. Patients generally prefer shared decision making, as it makes them feel empowered when choosing the treatment together with their physician [31,35]. Effective communication is important to facilitate shared decision making [33,35]. Since the BOS score is easy to interpret and is the objective result of a computer calculation, this can help in discussing the fracture risk and corresponding treatment plan together with the patient. Furthermore, being aware of the fracture risk can also be important for patients with respect to whether or not they have to adjust their physical activities.

Currently, the BOS score is only available for patients with metastases in the femur. We chose to focus on the proximal femur because it is often affected with bone metastases and fractures have a serious effect on the patients’ quality of life, for example, because the patients suffer from affected mobility and self-care. In the future, if the BOS score proves to be effective and of added value, we might develop finite element models for other bones as well. However, developing a new finite element model is a long process, as they need to be thoroughly validated ex vivo and in vivo.

This study has some limitations. As only physicians filled in the questionnaires and expressed their views of patient satisfaction, the patients’ point of view is anecdotal in the current study. Additionally, in the questionnaires, it was not specifically asked to indicate whether treatment plans were adapted or not. We estimated the changes in treatment plans based on the fracture risk prior to and after the BOS score based on the assumption that patients are treated in accordance with the clinical guidelines, namely, a single dose of radiotherapy in the case of a low fracture risk, and a higher dose of RT divided over multiple fractions or surgery in case of a high fracture risk [7]. However, it is possible that treatment plans were changed from, for example, surgery to multiple fraction RT due to the BOS score. Nevertheless, in 33% of the cases, it was specifically mentioned in the questionnaire that the treatment plan was adapted because of the BOS score. It should also be mentioned that it was not mandatory to complete the open-ended questions of the questionnaire; therefore, they were not always answered. In addition, some questions may not have been answered because the majority of the BOS scores were requested by the same physicians and they had already mentioned their comments in a previous questionnaire. For example, a physician’s answer to the question “are you satisfied with the BOS score” will probably not change when requesting multiple BOS scores. This could have caused underreporting of the results of the questionnaires.

Another limitation is that the BOS score is currently only applicable for patients with predominantly osteolytic metastases. Osteoblastic metastases show an increased density on the CT scan [36], which causes their strength to be overestimated by our FE model. In the future, we aim to select the regions containing osteoblastic metastatic tissue in the FE model using an automatic segmentation tool based on deep learning and change the corresponding mechanical properties to correct for the overestimation. Another limitation is that the complete femur needs to be visible on the CT scan to enable the alignment of the FE model to calculate the BOS score. In case of proximal femoral bone metastases, usually only the proximal part of the femur is scanned in clinical practice. Therefore, institutes have had to adapt their CT protocols in case a BOS score is requested. A new method for accurately aligning proximal femurs without the need of the complete femur would be very valuable and would enable the easier implementation of the BOS score on a larger scale. Another factor that complicates the wide implementation of the BOS score is the fact that we used a calibration phantom. We have developed an air–fat–muscle calibration method that enables calibration without a phantom [21,37]. However, the effect of differences between CT scanners on FE outcomes [38,39,40,41] in relation to air–fat–muscle calibration has not yet thoroughly been studied. Finally, the BOS score currently requires a protocolized CT scan used for radiotherapy planning. Other CT scans, such as diagnostic CT scans or PET CT scans, have not yet been used for BOS scores and should be validated before the wide implementation of the BOS score.

## 5. Conclusions

Based on these pilot results, we conclude that the BOS score for fracture risk assessment helps physicians and patients to make a more informed treatment-related decision on how to treat femoral bone metastases. In future studies, we will focus on investigating the value of the BOS score for shared decision making, and from the patient’s point of view. In addition, we aim to make the BOS score more accessible by developing a method to align femurs if only the proximal part is scanned, validate air–fat–muscle calibration, and increase the applicability of the BOS score to other CT scans.

## Figures and Tables

**Table 1 cancers-14-05904-t001:** Patient characteristics (*n* = 39).

Age in Years, Mean (Range)		65.3 (41–89)		
**Sex**, *n* (%)				
M		18 (46%)		
F		21 (54%)		
**Pain score**, mean (range) on a scale of 0–10 (0 = no pain to 10 = worst pain imaginable)		5.4 (0–9)		
**Karnofsky Performance Score (KPS)**, mean (range) on a scale of 0–100 (0 = dead to 100 = Normal; no complaints, no evidence of disease)		76.3 (60–100)		
**Primary tumor**, *n* (%)				
Breast		10 (26%)		
Lung		9 (23%)		
Prostate		7 (17%)		
Kidney		2 (3%)		
Colorectal		1 (2%)		
Multiple Myeloma		0 (0%)		
Melanoma		3 (8%)		
Other		7 (18%)		
**Type of bone metastases**, *n* (%)				
Lytic		31 (79%)		
Mixed		8 (21%)		
**Weight** in kg, mean (range)		79 (45–136)		
**Length** in cm, mean (range)		171 (150–196)		
**Patient reported quality of life**				
	**EQ-5D-3L ***, *n* (%)	**Level 1** **No problems**	**Level 2** **Some problems**	**Level 3** **Severe problems**
	Mobility	4 (10%)	33 (85%)	2 (5%)
	Selfcare	24 (61%)	12 (31%)	3 (8%)
	Usual activities	12 (31%)	19 (49%)	8 (20%)
	Pain and Discomfort	1 (3%)	27 (69%)	11 (28%)
	Anxiety and Depression	14 (36%)	23 (59%)	2 (5%)
	**EQ visual analogue scale (VAS) on patient’s self-rated health:** mean (range) on a scale of 0–100 (0 = worst health imaginable, 100 = best health imaginable)	60.8 (10–90)		

* The EQ-5D-3L (EuroQol 5 dimensions, 3 levels) [20] is a short questionnaire that is filled in by the patient to self-assess their health status based on five dimensions: mobility, self-care, usual activities, pain and discomfort, and anxiety and depression, each having 3 levels: no problems, some problems, and extreme problems.

**Table 2 cancers-14-05904-t002:** Fracture risk estimations by physician and based on BOS score, and the subsequent treatment decisions.

Risk Estimation Physician	Risk BOS Score	N	Fractures	No Treatment	Radiotherapy		Elective Surgery
					Single Dose	Multiple Fractions ^$^	
**low**	**low**	6		1	4	1	
**low**	**moderate**	7			1	6	
**low**	**high**	15	1		3	11 *	1
**high**	**low**	7			2	5	
**high**	**moderate**	2		1		1	
**high**	**high**	5	2				5 ^
Legend:	Fracture risk estimation of physician and BOS score are the same			Treatment probably adapted because of BOS score			Treatment may have been adapted because of BOS score ^#^

* = one fracture; ^ = two fractures shortly prior to elective surgery; ^$^ = Multiple fraction radiotherapy comprising 2 × 8 Gy, 5 × 4 Gy, 6 × 4 Gy, 10 × 3 Gy, and 13 × 3 Gy; ^#^ = In case of a moderate fracture risk indicated by the BOS score, it was difficult to assess whether treatment plans were adapted as we only have information about the initial estimation of the fracture risk, but not about the initial treatment plan. Additionally, in cases of patients who have a high fracture risk but do not undergo surgery, it is difficult to assess whether this is due to the BOS score or to other factors such as the patient’s wishes or general health status.

**Table 3 cancers-14-05904-t003:** Themes and findings from questionnaires that were filled in by physicians.

Theme	Narrative Answers	Number of Times Mentioned on the Separate Questionnaires * (*n* = 42)
Clarity of BOS score	The BOS score is a clear result	16
	The BOS score gives insight into fracture risk	11
	If the BOS score indicates moderate risk, this is difficult to interpret	3
	The BOS score is difficult to interpret if the weakest location of the bone is not clinically expected or is untreatable	2
Effect on decision of treatment	The BOS score caused a change in treatment plan	17
	The BOS score resulted in a well-substantiated treatment decision	10
	Due to the BOS score, multidisciplinary consultation with orthopedic surgeon was performed	8
Reassurance	The BOS score can give an extra confirmation of clinical estimation	16
	The BOS score can be reassuring and result in a feeling of security for the patient	8
	The BOS score can provide confirmation to the patient	6
Shared decision making	The BOS score helped to start a conversation between patient and physician	10

* It should be mentioned that the open-ended questions were not mandatory to fill out; hence, not all questions were always answered, probably because BOS scores were requested by the same physicians and they had already mentioned their comments in a previous questionnaire.

## Data Availability

The data presented in this study are available on request from the corresponding author. The data are not publicly available due to privacy and ethical restrictions.

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
