# Peer review of "A Patient-Specific Fracture Risk Assessment Tool for Femoral Bone Metastases: Using the Bone Strength (BOS) Score in Clinical Practice"

_cancers, 2022, doi:10.3390/cancers14235904_

Round 1

Reviewer 1 Report

Overall, this is an interesting study, however the authors need to describe some methods more and also moderate their statements about the efficiency of this test, as the data is not as strong as the what the authors say. The discussion is good and reports on limitations and references to other studies, which is appreciated. However more discussion needs to be added onto why the low results obtained in Table 3. For example, why does the BOS score not give a clear results nor insight on fracture risk (only 38% and 26%)?

Specific comments

1)     Isn’t the type of cancer something to consider helping the decision making, as it is known that some cancers are more lytic and others more blastic, or some mixed, and thus should be regarded as well?

2)     How does the FE model consider the type of mets? (referring now to lines 137-138 in 2.3), this should be added to the manuscript

3)     The methods mention osteolytic femoral bone metastases, It would be better to describe those type of fractures in the introduction, which may not been known to the general cancer audience. A quick description of what the differences are (and quality of life and treatment options, which are quite different) would add value.

4)     In 2.1, add the number of patients for each institute.

5)     Could the BOS score be done on other types of bones affected by metastases? It is true that the femur is highly afflicted zone, but other bones too, why was femur chosen?

6)     In the Supplmentary material: BOS Database, about ‘Based on this database, the sensitivity of the BOS score is 93%, the specificity is 72%, the PPV is 32% and the NPV is 99%.’ The methods describe calculation of the BOS score, but couldn’t find how ‘sensitivity’, ‘specificity’ and ‘NPV’ were measured. Please add in methods.

7)     Line 211, correct ‘Error! Reference source not found’

8)     Table 1 is very useful and interesting, the fact that blastic mets are not reported assumes that there were no cases of blastic only? Could the BOS score still be applicable for blastic mets only?

9)     Line 226, correct ‘Error! Reference source not found’

10)  ‘For 31 femurs, a discrepancy existed between the physician’s estimation and the BOS score’, this is confusing. It seems that there there are 39 patients, i.e. 39 femurs? In this case this would mean that in almost all cases the phyisican got a different estimation than the BOS? Is this not worrying? This also doesn’t match Table 2’s data.

11)  Line 279, correct ‘Error! Reference source not found’

12)  How do you explain that only 26% led to The BOS score giving insight in fracture risk

13)  Overall the results from Table 3 showed that the BOS score was not very helpful (less than 40% in any category), which is not sufficiently addressed by the authors in the manuscript

14)  The fonts of figures in pages 11 are too small

15)  What are the table and figure in page 10 and 11 belonging to? Can’t find a legend.

16)  Discussion: ‘The results of this pilot study indicated that the BOS score is of added value for physicians such as radiation oncologists and orthopedic surgeons because it is a clear score that provides objective insight in the fracture risk.’ Not realty based on table 3: The BOS score is a clear result: only 16 (38%) and the BOS score gives insight in fracture risk: only 11 (26%). The authors’ statements don’t support the actual results.

17)  ‘The diagnostic accuracy of the BOS score appears to be higher compared to other methods used in clinical practice (see supplementary material for current database)’. This is not visible from the supplementary material. And what are the ‘other methods used in clinical practice’? Be more precise.

18)  The authors discuss the limitations of the BOS score, which is important and good.

19)  ‘Another limitation is that the BOS score is currently only applicable for patients with predominantly lytic metastases. Osteoblastic metastases show an increased density on the CT scan,[34] which causes their strength to be overestimated by our FE model’. In this case, shouldn’t you also be careful of the mixed mets, which have a blastic component? How are you ensuring that the FE analysis of mixed lesions is representative? Wouldn’t it be better to exclude those patients?

Author Response

In the attachment, we answer all comments raised by the reviewer. 

Reviewer 2 Report

Dear Author,

This is an interesting paper. 

Here are my observations/questions/comments:

1.    Summary – Please explain abbreviation when first used (e.g. “CT”)

2.    Abstract – Instead of “at risk of fracturing”  I suggest “at high fracture risk” or “at high risk of fracture”

3.    Abstract – Please re-structure the Method section followed by “Results”

4.    Introduction – You should mention the year and the study at initial release of BOS score and, also, a brief introduction of the parameters that are taken into consideration to be clearer

5.    Methods – First section should be the study design followed by inclusion/exclusion criteria; then you should comment of prior studies/results on BOS score. Otherwise, it is not clear which is that study, which part is this study

6.    Results – Table 1- Did you take into consideration the specific site of metastasis at femur level?

7.    Results +  Table 2 – Did you take into consideration medical therapy like zoledronic acid or denosumab in addition or instead of radiotherapy?

Best regards,

Author Response

(The authors gave the same response as above.)
